# Only for Citizens? Local Political Engagement in Sweden and Inclusiveness of Terms

**Bozena Guziana** 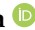

School of Business, Society and Engineering, Mälardalen University, 722 20 Västerås, Sweden; bozena.guziana@mdh.se

**Abstract:** In both policy and research, civic engagement and citizen participation are concepts commonly used as important dimensions of social sustainability. However, as migration is a global phenomenon of huge magnitude and complexity, citizen participation is incomplete without considering the political and ethical concerns about immigrants being citizens or non-citizens, or 'the others'. Although research on citizen participation has been a frequent topic in local government studies in Sweden, the inclusiveness and exclusiveness of terms used in the context of local political engagement, which are addressed in this article, has not received attention. This article examines the Swedish case by analyzing information provided by the Swedish Association of Local Authorities and by websites of all 290 municipalities as well terms used in selected research publications on local participation. Additionally, this article studies the effectiveness of municipal websites in providing information to their residents about how they can participate in local democracy. The results show that the term *citizen* is commonly and incorrectly used both by local authorities and the Association. The article concludes that the term *citizen* is a social construction of exclusiveness and the use of the term *citizen* should be avoided in political and civic engagement except for the limited topics that require formal citizenship.

**Keywords:** local political engagement; citizen; citizenship; resident; inclusiveness; exclusiveness

## 1. Introduction

"Democracy Day" is a yearly event organized in Sweden by Swedish Association of Local and Regional Authorities (SALAR). While attending this event in 2018, an elected official from the municipality of Vänersborg explained that the word citizen is outdated: "We have members of the council who are not Swedish citizens. There are those who feel excluded when the term 'citizen dialogue' is used instead of, for example, 'resident dialogue'. In my opinion 'dialogue' should be enough". This conversation triggered my interest in the inclusion–exclusion dimension of participation.

In the broad sustainability discourse, concepts such as 'place identity', 'physical and social integration', and 'participation' are common buzzwords signifying different, though often overlapping, targets for policy and research [1,2]. In this context of 'sustainabilities' and regardless of exact specification, local authorities are crucial actors stating their quest for healthy, equitable, and economically sustainable communities. Many local initiatives taken under the sustainability flag have a strong flavor of deliberation, communication, dialogue, and consensus, thus implying civic engagement is a key dimension in the implementation of sustainable development [3,4]. Thus, 'citizen participation'/'public participation' is commonly considered crucial for achieving 'social sustainability' in an urban/local context [5–7]. It is also argued that participation is important for successfully monitoring social development goals [8].

This article addresses the encouragement of public political engagement at the local level, including the inclusiveness aspects of terms used in targeting the public and for labeling instruments for participation. This article also provides comprehensive information

about different forms of participation. Sweden provides an interesting case because it is characterized by a high level of digitalization, a long tradition of development of citizen participation, and a high percentage of residents with foreign backgrounds.

Participation emphasizes the importance of citizens being active, not only at the time of elections but also in the intervals between elections. Researchers find that participation increases people's political self-confidence, their trust in the political system, and their understanding of the common good [9–14]. There is also a widespread agreement that including citizens will increase both the efficiency and legitimacy of government. Citizen participation is therefore loudly praised by decision-making authorities at all levels, national and local, and even the transnational level, such as the EU.

Parallel to the long-standing interest in citizen participation, the world-wide number of refugees and people in refugee-like situations is increasing exponentially. The UN Refugee Agency estimates that there were 80 million Forcibly Displaced People worldwide at mid-2020 [15]. Sweden has a long history of immigration. Recent immigration peaked in 2016 [16], which brought rapid changes in the population structure and especially to the growing number of residents who are not citizens in a legal sense. These changes have been noticed by the Swedish Contingency Agency (MSB).

During Emergency Preparedness Week in June 2018, the Swedish Contingency Agency sent out the brochure *If Crisis or War Comes—Important information for the population of Sweden* (Om krisen eller kriget kommer -Viktig information till Sveriges invånare) to all households in Sweden [17]. This brochure is now available online, with translations into three of Sweden's five minority languages (Finish, Meänkieli, and Sami) and into other languages such as Arabic, English, Farsi, French, and Russian, as well as a simplified Swedish version. The objective of this brochure was to prepare the people who live in Sweden for the consequences of anything from serious accidents such as extreme weather and IT attacks to—in the worst-case scenario—war. Similar communications were distributed in 1943, 1952, and 1961. Common to all these earlier editions was the reference to war, whereas the current edition (2018) states 'crisis or war'. More striking is the change in the terms used for the target group: people living in Sweden. In all earlier editions (1943, 1952, and 1961) *citizens* were explicitly addressed, while the edition from 2018 addresses *residents* in Sweden. This change in the term used in targeting people in Sweden is an example of using more inclusive language at the national level.

How are 'we' as people currently living in Sweden addressed in the context of political engagement and participation by authorities and by scholars at the local level? The present article argues that there is a need to discuss the use of the term 'citizen' as a crucial issue concerning who is included or excluded in a context when formal citizenship is not relevant. As many residents in Sweden's municipalities are not citizens in a legal sense, a growing percentage of constituents can be excluded by the terminology used by local governments on their websites.

Moreover, there are many different forms of citizen participation as a result of "participatory engineering [18] and the 'participatory revolution' [19]. Local authorities may increase public engagement by including an overview on their websites of the various participation tools that are available in their municipalities. As [20] (p. 25) pointed out, 'local political leaders in Sweden are the most supportive to party-based electoral democracy—and the most critical of participatory democracy—in Europe.' Therefore, it is especially interesting to investigate the comprehensiveness of information about opportunities for local political engagement provided on websites of local governments in Sweden.

Three following questions, not explicitly addressed by the body of literature, are posed in this article:

1. Are the local authorities encouraging political participation by giving comprehensive information about different tools for participation and influence?
2. Is the Association of Local Authorities and Regions taking leadership for adjusting democracy at the local level to the new reality in which an increasing number of residents in Sweden are not citizens in a formal sense?

3.  Are the local authorities using inclusive terms (resident) or exclusive terms (citizen) on their websites?

By studying these questions, the article contributes to the literature in three ways. Firstly, by focusing on the vocabulary used in democracy at the local level, the paper contributes to the literature on political engagement and social inclusion, especially regarding immigrants. Political participation is regarded as crucial for integration [21,22]. The use of inclusive or exclusive terminology in local democracy can influence political integration of non-citizen immigrants. Smith and Ingram [23] draw attention to ways that social groups can be constructed by policymakers in positive or negative terms. Clyne [24] highlights an important example of this language of exclusion that was used to divide the population of Australia into 'us' and 'them.' Lane [25] found a similar phenomenon in Norway, where the use of the exclusive term 'ethnically Norwegian' as a criterion of national identity led to 'heated debate' in media about the need for more inclusive identity categories suited for a multilingual and multicultural society. Recently, Barcena, Read, and Sedano [26] published their findings that show that inclusiveness of language in Language Massive Open Online Courses (LMOOC) of elementary Spanish for refugee migrants has a positive effect not only on migrants' language learning but even on social inclusion.

Secondly, the article contributes twofold to the literature about local digital democracy. Research shows that municipal websites can empower monitoring and participating in local governments [27] and that online information can also mobilize individuals for participation offline [28]. Notably, despite the growing number of e-democracy empirical studies, scholars and local leaders have shown little interest in the comprehensiveness of information on municipal websites that promotes both online and offline participation. The article fills this gap by studying the information on Swedish municipalities' websites. The high level of digitalization [29] and a long tradition of citizen participation make Sweden suitable for such study.

Thirdly, this article offers a novel methodology of using terms as indicators for inclusiveness. Within e-government research [30] and within practitioners' work [31] inclusiveness is often addressed as accessibility in the context of digitalization. This article broadens the view of digital inclusiveness in the time of growing migration by recognizing that minorities can be excluded by the information that is made accessible to them.

The remainder of this article is structured as follows. Section 2 describes the relationship between local governments' websites and democracy. Section 3 discusses citizenship and participation in a normative theory context. Section 4 outlines the development of citizenship regulation and lists instruments of participation in democracy at the local level in Sweden. Section 5 examines the Swedish case: information and materials provided by both the main actors within local participation, such as SALAR and local governments, as well as selected research publications on participation at the local level. Section 6 concludes that the term *citizen* is a social construction of exclusiveness and should not be used in the context of any public participation or civic engagement that does not require formal citizenship.

*Method and Material*

This article examines the inclusiveness of terms used by municipal webpages as well as information and material provided on the webpage of SALAR. Municipal governments and SALAR are Sweden's primary actors that facilitate political engagement at the local level. Additionally, this article reviews the terms used in selected research publications regarding local participation before and after Sweden's peak migration in 2016.

Digital inclusiveness is often addressed as accessibility, both within e-government research [30] and by practitioners [31]. This narrow understanding of accessibility is a response to the public nature of local authorities' responsibilities; '[u]nlike organizations in the private sector, government agencies have a charge to make their information and services available to everyone' [32] (p. 133). A growing number of residents in Swedish municipalities are not citizens in a legal sense. Therefore, this article studies another

dimension of inclusiveness: the terms that municipalities use to target individuals when facilitating information about political and civic engagement on their websites.

Municipality is the common legal label for the 290 local self-government units in Sweden, all of which currently provide information on their websites. At the time this study was conducted, the information on these websites was usually structured under 6–8 main headings, which correspond with public services operated by the municipalities such as schools, child and elderly care, utilities, housing, cultural and leisure activities, etc. Under these *main headings*, more information is available under a lot of *subheadings*, as shown in the Figure 1.

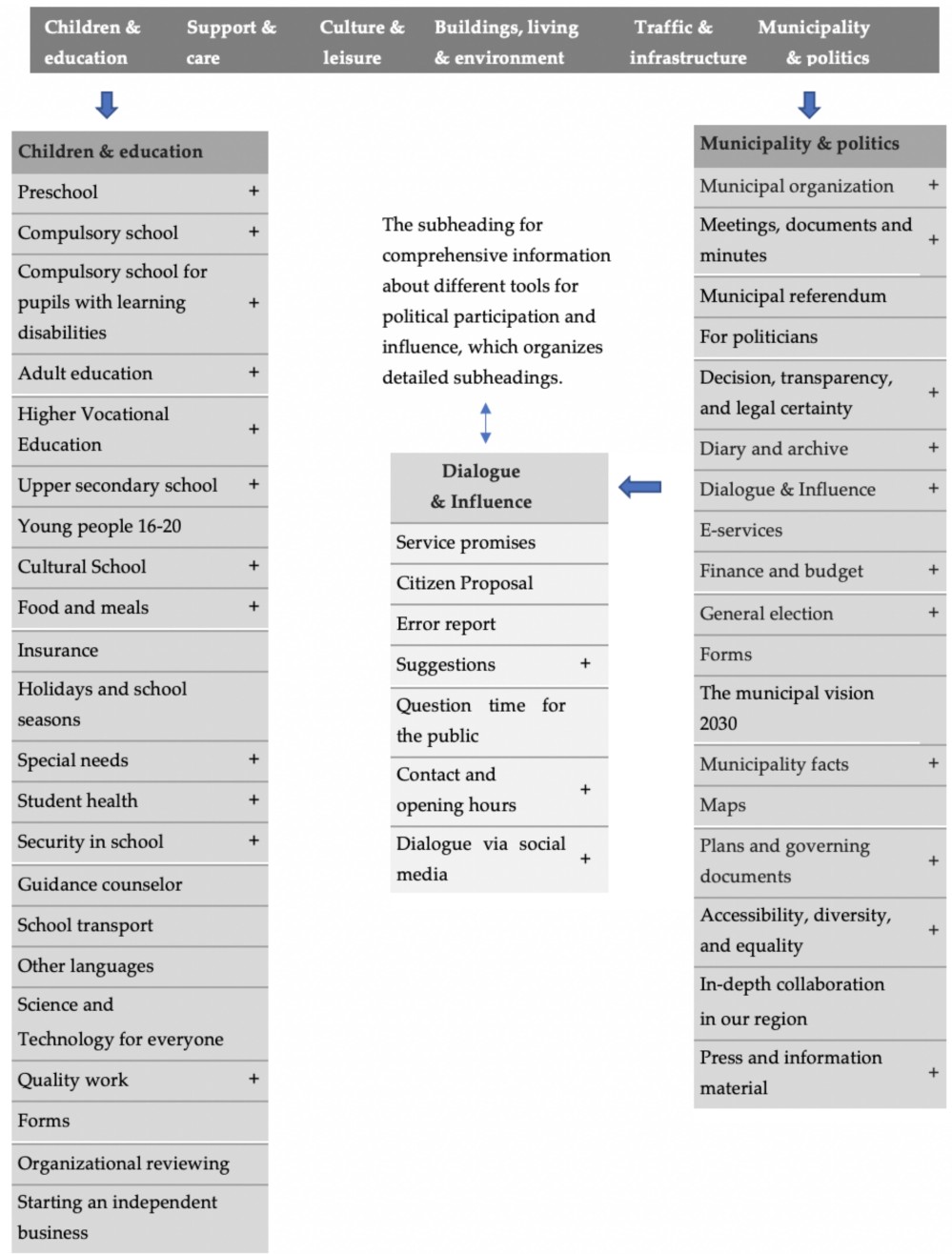

**Figure 1.** Example of main headings and subheadings on a municipal website.

Municipalities can make their information about means of participation easy to find by providing a comprehensive subheading, which organizes and displays the detailed

subheadings about specific opportunities for residents to participate. My findings below suggest that a percentage of municipalities have already achieved this level of accessibility.

Content analysis been conducted on all municipal government websites between 1 October 2017 and 6 January 2018 using an evaluation questionnaire (see Table 1).

**Table 1.** The factors for analysis of the municipal websites.

| Information about Participation | Inclusiveness of Language |
| --- | --- |
| The absence of a subheading for comprehensive information about different tools for political participation and influence at the local level under the subheading about the local government and politics (as shown in Figure 1). | The term used (citizen or resident): <br> * under websites' subheading for means of participation and influence <br> * when the authority offers dialogue as an opportunity for participation and influence. |

It is important to keep in mind the complexity of a municipality. A municipality can be seen as a geographic entity, an organization, and a political institution [33]. In relation to their residents, a municipality acts often as a service provider and as an authority. Residents are expected to meet public authorities and participate in politics, be taxpayers, voters, employees in municipalities, and users of the public activities that have expanded as Sweden's welfare state has evolved [34]. The information provided by the municipalities on their websites must meet this complexity. For example, to meet the growing interest among local authorities in the user role of the residents [35], the websites of Swedish municipalities commonly offer a function for accepting complaints as well as other suggestions about their services. The present paper focuses on the information related to the local political participation, i.e., the political agency of the residents. The user-related information and functions available on the municipal websites are therefore excluded.

## 2. Local Governments Websites and Democracy

Local communities and municipalities are crucial for the development and maintenance of democracy. They play an increasingly important role in our everyday lives. Research shows that people are closely connected to the local community and tend to be more interested in their own neighborhood or municipality than to their region or the whole country. Researchers also claim that the same pattern of local connection can be seen in the use of the internet [36]. 'Although the Internet is often seen as one of the big examples of McLuhan and Power's (1989) 'global village' concept, people using these global technologies often do this on a local level' [36] (p. 6). Many of the applications and information retrieval concern the local level.

The fast development of digitalization and e-governance is creating a growing interest in local authorities' websites. Earlier examples of studies have focused on more general questions such as *public involvement* [37], *e-participation* [38] and *e-government* [39]. More recently-conducted studies focus on more specialized issues, such as *technology acceptance* [40]; *local government transparency* (Portugal) [27]; *different determinants of adaptation* (Turkey) [41], (Norway) [42]; *information quality* [43]; *e-government evaluation models* (Greece) [44,45] and *use of social media,* for example, Italy and Spain [46], South Africa [47,48], and Western European municipalities [49].

In the beginning of the digital age, many scholars and practitioners thought that the new ICTs would contribute to democracy by connecting citizens with politicians and policy makers. This potential to enable citizens to communicate directly with government remains largely unrealized. The local authorities' websites tend to provide ample public information: contact information for public officials, descriptions of the activities of municipal departments, online council agenda minutes, and downloadable forms; thus, a kind of 'billboard' of information [28]. Here, the difference between *e-government* and *e-democracy* is relevant [50]. Generally, e-government deals with the passive provision of information and online services to individuals and businesses. By contrast, e-democracy

offers more active forms of public participation and engagement in decision making (for example [39,44,51]. Using this distinction, local authorities seem to be more interested in *e-government* than *e-democracy*; they primarily link the advantages of ICT with municipal service provision, for example, local authorities' websites provide information about services and self-service [52,53].

Steyaert [36] has studied information on Flemish municipalities' websites with regard to residents' different roles in the relationship to their municipality. Two of these roles involve political engagement: the role as a voter and the role as an active citizen. The third role is as a consumer or a client. The results of this study show that the municipalities tend to reduce the residents to a consumer or client of the services of the municipalities. 'The more political role of the resident, as a voter and especially as an active citizen, are not supported by the municipalities or even completely ignored' [36] (p. 15). It should be noted that Steyaert's research was published 20 years ago.

Municipal websites have changed since the beginning of the millennium. Now, Swedish municipalities, like municipalities in general, address residents in their political role with varying success. Still, population changes and new instruments for participation pose new challenges in addressing political engagement through local governments' websites. In the Swedish context, Lidén [54] has studied the supply of e-democracy on all municipalities using data between 2007 and 2009 as well as citizens' demand for e-democracy. The use of social media by municipalities has been studied by Klang and Nolin [55], Larsson [56], and Lidén and Larsson [57]. None of these previous studies has drawn attention to use of inclusive language on local government websites or to the comprehensiveness of the information these websites offer. The present article aims to fill these gaps.

### 3. Normative Theory Context: Citizenship and Participation

People have multiple social identities such as consumers, individual personalities, employees, members, and citizens. The role of being a citizen is confusing as citizenship is a widely contested concept (see [58] for a compilation). The literature presents different conceptualizations and dimensions of citizenship. Important examples include *civil, political,* and *social citizenship* [59]; *state* and *democratic citizenship* [60]; and, more recently, *digital citizenship* [61], *environmental citizenship* [62], and *local (urban) citizenship* [63].

Many studies explain the components of citizenship as (i) *legal status,* (ii) *political agency,* and (iii) *membership in a political community* [38,64,65].

When focusing on legal status and membership in a political community, citizenship is primarily a state-centered function; it includes national belonging with its associated rights and obligations. Britannica provides the following definition of citizenship [66]:

> [ . . . ] relationship between an individual and a state to which the individual owes allegiance and in turn is entitled to its protection. Citizenship implies the status of freedom with accompanying responsibilities. Citizens have certain rights, duties, and responsebilities that are denied or only partially extended to aliens and other noncitizens reside ng in a country. In general, full political rights, including the right to vote and to hold public office, are predicated upon citizenship. The usual responsibilities of citizenship are allegiance, taxation, and military service.

Accordingly, an individual can legally be a citizen of a state; individuals are not citizens of other administrative entities, such as municipalities, communities, or cities. Scholars identify two contradictory trends in the development of national citizenship: the ability of migrants to gain citizenship is becoming less challenging in some ways but also more challenging in other ways.

On the one hand, some political actors push for a liberalization of access to citizenship [67]. Examples include improved and simplified opportunities to obtain citizenship through registration; simplified naturalization rules, including the current period of residence; and recognition of multiple citizenship as some countries permit dual citizenship.

On the other hand, 'in the recent era of transnational population movements, national governments have harnessed both new and existing instruments to (re)assert state authority over the regulation of membership' [68] (p. 1153). Several states have considerably tightened access to citizenship and to permanent residence, and attitudes towards non-citizens have hardened [69]. For example, in the UK, the right of asylum seekers to receive the same benefits as settled citizens has been removed and replaced with reduced benefits [70].

While the path to citizenship is becoming less accessible, the value of citizenship is becoming more significant. As Joppke [67] pointed out, in the time of mass immigration, conflicts surrounding citizenship focus on the original meaning of citizenship as state membership. The tightening and loosing of citizenship rules are discussed by Mouritsen [71] in light of the literature on citizenship in sociology and political science representing 'post-national' critique and the rise of a global human rights regime [72]. The post-nationalization can be seen as a 'banalization' of the status of citizenship', and the new citizenship recognition discourse and policy can be seen as a "way of denying, resisting or reversing this post-national 'banalization'" [71] (p. 91). Furthermore, the 'banalization' of the status of citizenship is associated with two distinct developments [71]. One of them is the already mentioned liberalization of citizenship acquisition, the second one is the diminished importance of "material content and consequentiality of membership". Denizenship [73] is an example of the latter development, arguing that a state grants certain economic, social, and (sometimes) partial political rights to long-term residents who are settled within the state's borders but do not possess its citizenship. Elena Dingu-Kyrklund [74], in her paper on citizenship, migration, and social integration in Sweden, described that in the Swedish context, a resident enjoys almost the same rights as citizens in social, economic, and political terms, with some important exceptions. Thus "the citizenship issue, to a large extent, has been a secondary issue; the main and most difficult concern for non-Swedes remains that of immigration, which involves basic admission to and becoming officially domiciled in the country" [74] (p. 3).

The renewed significance of citizenship is illustrated by states attempting to differentiate more between the value of citizenship and mere residence, and in some countries between permanent and temporary residence. Hansen [75] stressed that the material and subjective value difference between citizenship and denizenship—permanent residence—in fact remains considerable and may be increasing. Ten years after Hansen's work, Hegelund [76] provided additional examples from Scandinavian countries of increasing differences between the rights of citizens and residents.

Moreover, not only *having citizenship* but *even having the right citizenship* becomes important. Research focused on global inequalities emphasizes the importance of location and its relationship with citizenship [77]. Citizenship and national location are the major factors behind differences in individual income across the globe [78,79], and the rise of a strategic-instrumental approach towards access to national citizenship [80,81] is not surprising. By strengthening the difference between citizens and non-citizens, the COVID-19 pandemic has influenced uses and meanings of citizenship in different ways, including creating problems and strategic choices for individuals who hold multiple citizenships.

In addition to the emerging importance of membership, which increases differences between citizens and non-citizen residents, nations also face the problem of the growing number of stateless people. Millions of people are stateless, and an estimated 1.1 billion lack legal identity documentation [82]. Since they are stateless, these individuals lack state acknowledgement and are 'often denied access to basic rights as education, healthcare, freedom of movement, and access to justice' [83] (p. 9). Scholars argue that territorial presence, not recognized national membership, should be the basis for migrants who are claiming rights [84,85] and should even be the basis for defining citizenship [86].

Citizenship is important for some forms of political agency. Voting and getting elected in national elections are examples of the privileges restricted to citizens. While adult citizens are entitled to vote in national elections even if they do not reside in the state, noncitizens may not vote in national elections even if they do reside in the territory of the

state. The decline in voting turnout in most advanced industrial countries [87,88] has led leaders and scholars to focus on activating citizens between elections. The concept of *citizen participation* has received much attention from different fields of study and is loudly praised by decision-making authorities (see Bobbio [89] for an overview of different arrangements for participation). There are various definitions of citizen participation. Verba, Scholzman, and Brady [90] defined it as any voluntary action by citizens that is more or less directly aimed at influencing the management of collective affairs and public decision making. Arnstein [91] introduced a ladder of participation, from elemental to more in-depth participation (e.g., information, communication, consultation, deliberation, and decision making) based on levels of interaction and influence in the decision-making process. Swedish scholars have contributed to understanding the governing of participatory instruments through studies of, among others, participants' motives for participation [92], idealist and cynical perspectives on the politics of citizen dialogues [93], invited participation under pressure in a local planning conflict [94], and the rise of e-participation initiatives in non-democracies [95]. In addition, Hertting and Kugelberg [96] shed light on the problem of institutionalizing local participatory governance in relation to representative democracy.

The broad attention for strengthening participation is described by scholars as 'participatory engineering' [18] and 'participatory revolution' [19]. Besides the meaning of membership of a state, the term *citizen* is therefore commonly used for the political agency in the context of political and civic engagement; some scholars are even pointing out 'citizenship turn': '[w]hatever the problem—be it a decline in voting, increasing numbers of teenage pregnancies, or climate change—someone has canvassed the revitalization of citizenship as part of the solution' [97].

When most residents in countries were also citizens in a legal sense, the use of the terms citizen and *citizen participation* was rather unproblematic. Now, however, no part of the Earth is unaffected by migration, either as a country of origin or as a country of destination. Therefore, when scholars or government authorities use the term *citizen* to imply political beings with certain rights and obligations, this should raise serious concerns.

## 4. Citizenship and Local Participation in Sweden

### 4.1. Citizenship

For a long time, Sweden has been counted as a state with one of the more inclusive citizenship regulations [74,98]. This legislation has evolved over time in close cooperation with other Nordic countries; it started at the end of the 19th century and continued after the Second World War. For example, these joint discussions on citizenship led to adaptation of new citizenship laws in Denmark, Norway, and Sweden in 1950 [98].

Both EU-integration and globalization influenced a number of modifications that were made to Swedish legislation; among others, the positions within the Swedish bureaucracy and judiciary that should be restricted to citizens were reduced to very few positions. Only some top positions in the judicial system (judge, prosecutor) and some few top administrative positions still require Swedish citizenship. Others, such as ordinary judicial and related positions (lawyers, jurors) that previously had required Swedish citizenship, are now no longer restricted. This inclusiveness towards non-citizen residents separated Sweden from other Nordic countries; the revised Citizenship Act, adopted in 2001, was the first codification that was not based on Nordic cooperation [74]. The liberalization of citizenship rules has been more far-reaching in Sweden than in other Scandinavian countries [98]. For example, acceptance of dual citizenship was introduced in Sweden almost two decades before Norway [99]. The main landmarks in the development of Swedish citizenship legislation are presented in Table 2. The Swedish legislative process begins with the appointment of a specialized committee that is given a mandate to investigate the question to be legislated. This committee produces a report with a suggestion for new legislation, a so-called SOU or Ds. SOU—Sveriges Offentliga Utredningar (Sweden's Public Investigations); Ds—Departements serien (Ministry series, a smaller, shorter form of public investigations handled within the ministry in question) [74].

**Table 2.** Development of legislation of Swedish citizenship with a focus on acquisition of citizenship and participation. Partly based on [74,98].

| Year | Legislation and/or Committee Report | Effect on Acquisition of Citizenship and Participation in Sweden |
|---|---|---|
| 1894 | Citizenship Act | |
| 1924 | Citizenship Act | New naturalization conditions: (1) be over 21 years of age, (2) have accumulated 5 years of domicile in Sweden, (3) have exhibited good conduct, and (4) demonstrated a capability to provide for himself and his family. |
| 1950 | The 1950 Act (1950: 382) on Swedish citizenship | Established common immigration policies across Scandinavia that allowed selected rights to immigrants. |
| 1976 | SOU 1975:15 [100] Municipal suffrage for immigrants | The electoral reform gave electoral rights to foreign citizens who had lived at least 3 years in Sweden. The three-year waiting period for citizens from EU countries, Iceland, and Norway was abolished [101]. |
| 1976 | | A major reform of the naturalization rules in Sweden: the waiting time for Naturalization was shortened from 7 to 5 years, and to 2 years for Nordic citizens. The residence request for acquisition by notification was shortened from 10 to 5 years. |
| 2001 | Revised Swedish Citizenship Act (2001: 82) SOU 1999:34 [102] Swedish Citizenship | Acceptance of dual citizenship |
| 2013 | SOU 2013: 29 [103] Swedish citizenship | The definition of the meaning of Swedish citizenship: 'Swedish citizenship is the most important legal relationship between the citizen and the state. Citizenship involves freedoms, rights and obligations. It is a basis for Swedish democracy and represents a significant link with Sweden.' Annual Citizenship ceremonies for new Swedish citizens should be held by municipalities. |
| 2021 | SOU 2021:2 [104] Requirements or knowledge of Swedish and social studies for Swedish citizenship | Pending |

Swedish regulations regarding citizenship ensure that a resident enjoys almost the same rights as a citizen—in social, economic, and political terms [74]. Two notable exceptions are: '[ ... ] the right to vote in national elections and, especially the unrestrained, inalienable right to reside, which are still exclusively reserved for citizens' (p. 8).

Residents with a foreign background represent a growing share of the population in Sweden. Similarly, the number of applications for citizenship is also growing, as shown in Table 3. Foreign background includes foreign-born and domestic-born with two foreign-born parents. Before 2001, even domestic-born with one foreign-born parents was considered as foreign background [105].

**Table 3.** Granted application on citizenship [106].

| Year | Granted Applications | Share of Population with a Foreign Background (%) |
|------|---------------------|--------------------------------------------------|
| 2000 | data not available | 14.5 |
| 2010 | 28,100 | 19.1 |
| 2011 | 33,112 | 19.6 |
| 2012 | 46,377 | 20.1 |
| 2013 | 46,849 | 20.7 |
| 2014 | 38,890 | 23.5 |
| 2015 | 44,209 | 22.2 |
| 2016 | 56,037 | 23.2 |
| 2017 | 65,611 | 24.1 |
| 2018 | 61,309 (* 93,261) | 24.9 |
| 2019 | 74,924 (* 109,580) | 25.5 |
| 2020 | 81,377 (* 119,728) | data not available |

* The number of submitted applications.

Unlike many others European countries, Sweden at present has no language or civics tests for people applying for citizenship. There is, however, a good conduct clause requirement, and either a criminal record or unpaid debts can affect applications. Furthermore, applicants need to have lived in Sweden for five years, or three if they are a cohabiting partner of a Swedish citizen, before they can apply for citizenship. However, the Swedish government has launched an inquiry that investigates how the law could be changed to make it compulsory for applicants to pass a test on the Swedish language and civics to get citizenship. The final report is to be presented by 1 July 2021, with the parts of the report dealing with the language and civics tests presented 13 January 2021 [103]. The Inquiry states that the purpose of the requirement is to enhance the status of citizenship and promote an inclusive society. The Inquiry proposes that the Swedish Citizenship Act (2001:82) should stipulate that knowledge of Swedish and civics is required for the acquisition of Swedish citizenship. The knowledge requirement for Swedish citizenship should cover people who have turned 16 but not 67 years. However, this requirement should not apply to state-less persons born in Sweden who are under 21 or to Nordic citizens who acquire citizenship through the provisions on notification in Section 18 and Section 19 of the Citizenship Act. The citizenship test in civics should be based on a book specially produced for the purpose. The book should contain knowledge needed to live and function in Swedish society focusing on democracy and the democratic process and it should be available for download in Swedish and ten immigrant languages in Sweden. Additionally, the government is also looking into introducing a similar requirement for obtaining permanent residence. In sum, Swedish citizenship laws include both contradictory trends: facilitating access to citizenship while also restricting citizenship.

### 4.2. An Overview of Forms for Participation

The Swedish constitution states that public power in Sweden is derived from the people and there are three levels of domestic government: national, regional, and local. Sweden is divided into 290 municipalities and 21 regions. In January 2020, the county councils (landsting) of Sweden were officially reclassified as Regions (regioner). In addition, there is the European level, which has acquired increasing importance following Sweden's entry into the EU.

The 1992 Swedish Local Government Act (LGA) regulated the division into municipalities and county councils as well as the organization and powers of these municipalities and county councils. The LGA states that all residents are *members* (a person who is registered as a resident of a municipality, owns real property there, or is assessed for local income

tax there is a member of that municipality [107]) in a municipality, not *citizens*. With the new 2017 LGA, the county councils were changed into regional councils. The Swedish Association of Local Authorities and Regions changed the name in Swedish (from Sveriges Kommuner och Landsting (SKL) to Sveriges Kommuner och Regioner (SKR); n English the name is SALAR, in both cases), and the county councils (landsting) of Sweden were officially reclassified as Regions (regioner) in 2019. The LGA also contains rules for elected representatives, municipal councils, executive boards, and committees. There is no hierarchical relationship between the local and the regional level since municipalities and regions have their own self-governing authorities with responsibility for different activities.

In Sweden, like most countries, *voting* is a political act, limited at the national level to members of the state. However, as it is shown in Table 2, since 1976 foreign citizens who had lived at least 3 years in Sweden had electoral rights in municipal elections and in elections to the county council assembly. This three-year waiting period was abolished for citizens from EU countries, Iceland, and Norway in 1998 [104]. Bevelander and Spång [108] refer to EUDO-Citizenship Observatory 2016 and emphasize that Sweden, Denmark, and Finland are the most inclusive countries in Europe when it comes to voting rights for non-EU citizens). Since 1970, elections for local and regional level representatives have been held on the same day as the general election in Sweden.

The opportunity for public participation in planning and building processes has a long-standing tradition and is obligatory and meticulously regulated in Sweden and other Nordic countries [109]. However, this paper has been exclusively focused on other forms of participation. It can be added that in the Swedish Planning and Building Act the term 'citizen' is not used, only 'resident' (boende).

In the beginning of this millennium, researchers and policy makers promoted new initiatives to activate citizens between elections in Sweden [110] partly due to recommendations from the first Commission on democracy. The final report of the Commission, 'Sustainable Democracy. Policy for the Government by the People in the 2000s' included a number of proposals concerning local democracy and also stated that 'every citizen must be afforded greater opportunities for participation, influence and involvement.' [111] (p. 243). Furthermore, the Government Democracy Bill from 2001 [112] declared democracy as a policy area of its own and encouraged the 'municipalization of democracy' [113] (p. 136). The need for democratic expertise emerged along with the view that democracy is an issue not only for political parties but also for the municipalities. A growing number of municipal officials, 'democracy operators' [113] (p. 87) work to promote local democracy with support from SALAR.

Table 4 depicts an overview of the formal instruments of participation in democracy at the local level in Sweden. *Voting, contacting politicians* and *attending public questions time at council meetings* are examples of traditional political acts at the local level. *Citizen dialogue* and *e-petition* are examples of tools for participation. In 2006, the Swedish Association of Local Authorities and Regions initiated a project to support both citizen dialogue and e-petition as vital tools for civic engagement [114]. This Association has actively promoted the citizen dialogue by producing a great deal of published information, working with networks, conferences and awards, etc., thus acting as a 'policy entrepreneur' [93]. In 2015, 83 percent of Sweden's municipalities and county councils (stated that they had implemented some form of citizen dialogue [115].

**Table 4.** Examples of formal instruments of participation in democracy at the local level in Sweden.

| Instrument | Requirements |
|---|---|
| National election | Swedish citizen aged 18 or more, who is or has been registered in Sweden |
| Voting (local election) | Citizens of other EU countries, Iceland, or Norway who are registered in the municipality or county Citizens of other countries who have been registered in Sweden for a minimum of three years and are registered in the municipality |
| Contacting politicians or local government officials | none |
| Public question time at a council meeting | none |
| Participation in citizen dialogue/resident dialogue (medborgardialog/invånardialog) | none |
| Proposing or signing a people's initiative (folkinitiativ) (instrument for direct democracy) Introduced 1994, strengthened 2011 | Residents with voting rights in the local election |
| Submitting a "citizen proposal" ((medborgarförslag) * Introduced 2002 This instrument has been regulated in LGA, which gives it a special position among other participatory instruments) | Members of the municipality, i.e., registered in the municipality |
| Proposing or signing an e-petition * | Varied; residents (Västerås), registered members (Haninge), everyone (Borås) |

* depending on the availability of the instrument.

A common goal among the different forms of participation is activating voters between elections. Another goal is to involve in local politics those without a legal right to vote, such as children or residents with foreign backgrounds. In 2002, an instrument called *citizen proposal* was introduced to target both goals. For local authorities that decided to implement this instrument, the *citizen proposal* (CP) enables all residents who are registered in a municipality, including those without a legal right to vote, to raise issues to the local government regarding local areas of responsibility. The CP process was included in the Local Government Act (LGA) on 1 July 2002, which gives this tool a special position among participatory instruments. Other instruments, such as e-petitions and citizen dialogues, produce suggestions that local representatives are not obliged to take into account. According to the LGA, the *citizen proposal* should be processed to enable the council to make a decision within one year of the date on which the citizen proposal was tabled, and the processing of the CP should be described in the standing order for the assembly. This instrument has spread to more than half of Swedish municipalities. In 2016, 188 (65%) of Sweden's 290 municipalities had information on their websites about how citizens can submit a citizen proposal [116].

Instruments for *direct democracy* are relatively weak in Sweden. Although Sweden has been ranked as the 'most democratic country' in the world [117], there are limited formal rights for direct civic participation [118]) and Sweden is instead characterized by a lack of 'referendum culture' [119]. For example, since introduction in 1921 only 6 national referendums have been held. Since 1977, institutionally initiated referendums have been allowed to be held at local and regional levels [119], which have been regulated in the LGA. The regulation of local and regional referendums has been changed twice during this time. Due to inspiration from Finland, the *people's initiative* has been introduced with the constitutional amendment of 1994 and in the new Municipal Referenda Act [120]]. The *people's* initiative gives residents with voting rights the ability to initiate a referendum

process by getting 5% of the population in a municipality to sign a petition. However, the referendum would only be enacted if a majority of the municipal assembly approved of the referendum. In practice, very few initiatives were forwarded by the local authorities to the electorate for a consultative popular vote. The dysfunctionality of this tool led Sweden to strengthen the *people's initiative* with the constitutional amendment of 2011: only if 2/3 of a municipal assembly opposed any specific people's initiative could the referendum be denied. This has led to several policy changes initiated by people referendums [121]. By 2018, 174 referendums initiated by people in 105 of 290 municipalities have been enacted [122], i.e., the majority of municipalities have not yet achieved this form of participation.

In various international rankings, Sweden appears among the most advanced OECD countries in terms of the level of digitalization of its society and economy. In the Digital Economy and Society Index 2018, Sweden 'ranks second regarding the use of Internet by its citizens, third in terms of the use of Internet for transactional services (including banking and shopping), and third in terms of individuals' use of the Internet to send filled forms to public authorities' [29]. As there are different forms of participation as well as other online options for giving suggestions to the municipality, local leaders can simplify participation for their residents by providing an overview of participation tools using inclusive language. Local government websites that provide clear, inclusive information about participation can help residents to get involved in local politics. These aspects of the online information of the municipal websites are studied in the present paper.

## 5. Terminology in the Swedish Case

### 5.1. Previous Research

As participation at the local level has gained attention from politicians and practitioners, different research projects in Sweden have studied participation. This section provides a review of the terms used in examples of research publications before and after 2015 (see Table 5). The year 2015 'was characterised by very strongly increasing numbers of people seeking protection. In total, Sweden registered almost 163,000 new asylum applicants, more than twice as many as during the year before, which had already marked a record' [123] (p. 4). In all the studied publications, both the term 'resident' and 'citizen' are used.

Two of these publications have 'residents' or 'residents dialogue' in the title. One of them, 'The future is already here. How residents can become co-creators in the city's development' [124] is a book published by a research project about the interplay between citizen initiatives (medborgarinitiativ) and invited participation in urban planning. Despite including 'residents' in the title, most of chapters of this publication use the term *citizen* as well as *citizenship* for the political agency. However, in two chapters written by Stenberg [124] the term *resident* is used, except in the general expression 'citizens role in the planning' in the title of one of these chapters, and the introduction to the second chapter.

The other publication, published within a project about justice and socially sustainable cities, is titled 'The role and forms of resident dialogue The Västra Götaland region's consultation with civil society' [125]. In this publication, the term 'resident dialogue' as well as 'resident' are used considerably more frequently than 'citizen dialogue' and 'citizen', but still not consistently and without any reflection on differences between these terms. In a previous publication in the same project [126], the use of the term *citizen dialogue* (30) in comparison to the term *resident dialogue* (3) was dominant. This indicates a purposeful change in the choice of the vocabulary in the publication from 2015. The lack of discussion on the differences between terms used in this publication is therefore even more striking.

In spite of those examples indicating some awareness of the terms used, an explicit discussion of the inclusiveness/exclusiveness of the terms is lacking. Only two short footnotes are included that address terminology: one footnote states that *citizens* and *citizen dialogue* 'should not be understood in a narrow legal sense but as synonymous with those who live, stay and work in the city' [127] (p. 4); and the second footnote highlights 'the linguistic challenge of using the term *citizen* in the time when more and more people are living as non-citizens' [128] (p. 77).

Notably, there are no considerations on inclusiveness of language in the newly published special issue of the Swedish planning journal Plan [128] on 'Planning and democracy'. The contributions by established scholars on participation and local governance do not draw attention to resident terminology. 'In PLAN, researchers and practitioners describe, analyze and debate changing conditions and new challenges for community planning, new working methods and the development of the profession. PLAN monitors municipal and regional planning, regional policy, construction and housing policy, infrastructure, environmental policy and international development trends, and highlights social consequences in planning. PLAN makes room for new theoretical perspectives and young writers!' [129].

By contrast, Wiberg [130] in her article 'The political organization of citizen dialogue' published at Stockholm center for organizational research (SCORE) mentioned the exclusiveness of the term *citizen dialogue* and inclusiveness of the term *resident dialogue* as not all residents are Swedish citizens. Wiberg also pointed out that several municipalities (p. 7) have started to use the term *resident dialogue* instead of *citizen dialogue*, e.g., Halmstad municipality.

**Table 5.** Terms used in selected examples of publications on local public participation.

| | Stenberg et al., 2013 [124] | Olofsson, 2015 [127] | Abrahamsson et al., 2015 [125] | Jahnke et al., 2018 [131] | Plan 2021 [128] |
|---|---|---|---|---|---|
| Title | 'The future is already here. How *residents* can become co-creators in the city's development' | 'A research-based essay on the possibilities and obstacles of dialogue' | 'The role and forms of *resident dialogue* The Västra Götaland region's consultation with civil society' | 'Management system for better *citizen dialogue*. A study of the management and organization of dialogue work in the public construction sector' | 'Planning and democracy' |
| Resident * | 60 | 27 | 84 | 10 | 14 |
| Citizen * | 253 | 26 | 17 | 1 | 40 |
| Resident dialogue | 1 | 1 (references) | 26 | - | - |
| Citizen dialogue | 73 | 36 | 7 | 21 | 117 |
| Reflection about terms used | – | Footnote in Summary: Citizens and citizen dialogue should not be understood here in a narrow legal sense but as synonymous with 'those who live, stay and work in the city' (p. 4) | – | – | Footnote in chapter by Ingemar Elander: Even the word 'citizen' is a linguistic challenge today when more and more people are living as non-citizens within the national border, they happen to be in. 'Refugees', 'asylum seekers', 'undocumented', 'unaccompanied minors', 'illegal', 'irregular', 'the others', 'nomads', and 'invisible' are some of the different names. (p. 77) |

| | Stenberg et al., 2013 [124] | Olofsson, 2015 [127] | Abrahamsson et al., 2015 [125] | Jahnke et al., 2018 [131] | Plan 2021 [128] |
|---|---|---|---|---|---|
| Examples of the use of the term *resident* | * resident initiative (p. 95) | * make residents co-creators of societal change (p. 18) <br> * resource-poor residents (p. 28) | * city residents (p. 21) <br> * dialogue with residents (p. 30) | * resident involvement (p. 6) | * Citizen participation is not just about giving residents the opportunity to participate in digital platforms (p. 84) <br> * dialogue between residents and planners (p. 104) |
| Examples of the use of the term citizen | * citizen initiative (p. 81) <br> * democratic citizen participation (p. 100) <br> * citizen partici-pation (p. 108) | * a politically active citizen between the elections (p. 11) <br> * responsible citizens (p. 40) <br> * the empowered citizen (p. 40) | * citizen democratic influence (p. 31) <br> * low-abiding citizens (p. 9) | * citizen participation (p. 4) | * in our role as a citizen (p. 105) <br> * dialogues between municipality, citizens and builders (p. 25) <br> * concerned citizens (p. 23) |
| Other terms | * people (p. 23) | * People's political participation (p. 112) | * persons participation (p. 19) | * different inndividuals (p. 6) | * people (p. 8) <br> * people's (p. 3) |

### 5.2. The Swedish Association of Local Authorities and Regions (SALAR)

The mission of SALAR is "to provide municipalities, county councils and regions with better conditions for local and regional self-government. The vision is to develop the welfare system and its services. It's a matter of democracy." The Association is involved in promoting local democracy in Sweden, often acting as a 'policy entrepreneur' [92].

*Democracy, leadership, governance* (Ddemokrati, ledning, styrning) is one of the eight main headings on SALAR's website. As shown in Figure 2 SALAR uses the term *citizen*, even when targeting people who wanted more information about the LGA, initiatives, and referendums. In its publication on the strengthened 'people's initiative', SALAR refers to 'citizens entitled to vote in the municipality or county council') 'röstberättigade medborgarna i kommunen eller landstinget.') [132] (p. 2). The Association uses this *citizen* term despite the aforementioned fact that citizenship is not required in order to vote in local elections. Similarly, in a platform for discussion and standard setting "Ten factors for 'good local democracy'", SALAR explains that citizen participation is one of the factors, thus referring to municipal residents as *citizens*.

**Figure 2.** Targeting people who need or want more information about the LGA and instruments for local participation (own translation: Citizens in Sweden who have questions about the Local Government Act, people's initiative, referendum).

In their efforts to support municipalities and regions, SALAR conducts different surveys and comparisons. Some surveys are related to democracy issues such as the Democracy Barometer (Demokratibarometer) and Information for All (Information för alla). The survey Information for All was conducted yearly between 2009 and 2017, with around 250 questions regarding different municipal services (preschool, elementary school, high school, elderly care, individual and family care, disability care, building and living, streets, roads and environment, permits, business and more, non-profit sector, culture and leisure, and business), as well as transparency and influence, and municipal websites' search functions. SALAR reports that these surveys have 'significantly contributed to developing websites of the municipalities based on citizen perspectives' and 'the results have improved year by year' [31] [p. 5]. The majority of questions asked in Transparency and Influence (see Appendix A) are about what information the website users can access; there are few questions about how the information is available or for whom (audio, sign language, other languages, web TV, etc.). There are no questions about websites' description of the means for political engagement and participation. SALAR also conducts studies on various issues, including participation and democracy, among others. Similarly, SALAR's study about the people's initiative [121,133] focused on whether websites' users could find information about specific people's initiatives that are in progress in the municipality; this study did not ask if websites explain that residents have a right to initiate a new referendum.

*5.3. Local Government Websites*

The entire territory of Sweden is divided into 290 local self-government units. All of these local units provide information on their websites. The results of content analysis of these websites are presented below.

5.3.1. Main Subheading for Participation and Influence

A vast majority of municipalities' websites include some heading related to local politics, often *Municipality and Politics* (*Kommun and politik*). Surprisingly, only 71 precent of municipalities provide a comprehensive subheading for the means of participation under their heading related to local politics. Table 6 depicts the topics in these subheadings that encourage residents to engage in local politics. 'Influence' and 'dialogue' are most frequently used. The choice of the term 'influence' can be explained by its explicit correspondence to agency. A possible explanation for the term 'dialogue' can be the strong establishment of 'citizen dialogue' as an instrument for participation. Notably, some municipalities choose to combine participation and making complaints about their services.

**Table 6.** Topics in subheadings for comprehensive information for participation and influence on municipal websites.

| Topic | | Share |
|---|---|---|
| Influence | (Påverka) | 43% |
| Dialogue | (Dialog) | 36% |
| Democracy | (Demokrati) | 8% |
| Insight Access | (Insyn) | 5% |
| Appealing a decision | (Överklaga) | 1% |
| Others | | 6% |

Under the subheading for comprehensive information, the local governments list different forms of participation. Which forms are listed varies. The variation partly depends on the decision by the local authority to introduce instruments such as citizen proposals or e-petition. Very few, only 5%, of all municipalities provide information about their residents' right to use the 'people's initiative'.

In some few cases, municipal websites organize the online content under this comprehensive subheading by roles instead of actions; for example, there are subheadings for

councilors (fullmäktigeledamot), elected officials (förtroendevald), citizens (kommunmedborgare), and municipality residents (Hudiksval).

### 5.3.2. Targeting Residents

Municipalities can activate their residents by explicitly including residents and participation options on their municipal websites. As such, the analysis presented below includes two aspects of targeting residents on local websites. The first aspect focuses on how residents are targeted under the subheading for comprehensive information about participations tools. The second aspect focuses on what term is used when the authority offers dialogue as an opportunity for participation.

#### Under Subheading for Participation and Influence

As shown in Table 7 the term 'citizen' is used by 32 percent of municipalities. Some communities are addressing their residents as 'You a citizen' for example 'You a citizen in the city of Gothenburg'. Other communities use the term 'municipal citizen' (kommunmedborgare) (e.g., Håbo, Bengtsfors).

**Table 7.** Terms used to address residents.

| Terms | | Share |
| --- | --- | --- |
| Citizen | (medborgare, kommunmedborgare) | 26% |
| You | (du) | 22% |
| Resident | (invånare, kommuninvånare) | 21% |
| Mixed (both resident and citizen) | | 6% |
| None | | 24% |

#### Citizen or Resident Dialogue

The term *citizen dialogue* (medborgardialog) dominates under the subheading for information about participation and influence or the main heading for local politics; only 8 municipalities use the subheading *resident dialogue* (invånardialog). Moreover, some municipalities also published a steering (policy) document on the process of citizen/resident dialogue (Bollebygd, Forshaga). Notably, in some few cases, such a document is offered even when a clear subheading for the dialogue as an instrument for participation is missing (Västerås stad, Lilla Edet, Vänersborg). The publishing of these policy documents can be the result of the SALAR's efforts to support municipalities with development of citizen dialogue as instrument for participation, where the local authorities focus on development of steering document for the use of municipal officials but fail to provide clear information for the residents.

### 6. Discussion: Local Political and Civic Engagement for Citizens or for Residents?

*Citizenship* is a political concept familiar to the majority of people that connotes membership in a particular country. The term *citizen* is also well established in the context of participation as *citizen participation*. Norris [134] has developed the concept of 'critical citizens' to describe the long-term trend of people becoming more critical to the political systems. In times when immigration and multiculturalism pose new questions about citizenship, it is time to be critical about the use of the term 'citizen' in political and civic engagement. National elections continue to require a legal distinction between citizens and non-citizen residents, but many other forms of participation do not require this divisive distinction. Citizenship is a term that is too narrow and exclusive: populations are changing, and hardening boundaries between citizen and non-citizens are contributing to growing exclusiveness in the terms *citizenship* and *citizen.* Kuwait provides an extreme example of the importance of replacing the term "citizen" with more inclusive language, as citizens only comprise 32% of the population. Mirchandani, Hayes, Kathawala, and Chawla [135]

use the term "resident" instead of "citizen" in their article on preferences for e-government services and portal factors. As Bosniak [84] and Ochoa Espejo [85] pointed out, territorial presence should be more central than national membership, i.e., people should be seen as residents, not as citizens.

The use of the term *citizen participation* and *citizen* for political agency, as established both among scholars and local authorities, is no longer appropriate. Civic participation problems due to citizenship are stressed by the United Nations Department of Economic and Social Affairs [136] (p. 47) in their report about social inclusion:

> In diversified local environment, it is easier for many residents to identify with the city where they live, work and interact, rather than the national state. In policy discourse related to social inclusion, citizenship is frequently invoked . . .

> Citizenship, by definition, is membership of a political community and includes rights to political participation. There is a need for finding a way to allow inclusion of all residents in a particular location so that none of them are excluded or marginalized . . . A new construct of "membership" in cities may be considered as a solution.

These problems concern formal issues related to citizenship, but as the citation in the beginning of this article shows, and as pointed out by Clyne [24], words matter; language is either more inclusive or less inclusive. Some local authorities in Sweden have taken positive steps to switch their terminology from 'citizen' to 'resident' as in the case of 'resident dialogue', which is also observed by other scholars [130]. The Swedish Contingency Agency (MSB) has provided an example of inclusive language on the national level [17]. Consequently, SALAR's use of 'citizen dialogue (including publication in English (SALAR, n.d.)) and addressing residents in Sweden as 'citizens' in the context of local democracy is therefore even more astonishing.

Moreover, it is problematic that some instruments for increased political participation at the local level are labelled with 'citizenship'-related terminology, such as 'citizen dialogue' or 'citizen proposals'. Sweden introduced citizen proposals to involve those without a legal right to vote, such as children or residents with foreign backgrounds. Municipalities are using the term 'citizen proposal' and then explaining in the information about the instrument who can submit such proposals; often as 'you who are registered in' (e.g., Haparanda). However, some municipalities also tend to use the term 'citizen' in the description of the instrument, as shown in the following examples: 'Citizens, i.e., those who are listed in the municipality, can submit ('medborgare, dvs de som är folkbokförda i kommunen, kan lämna förslag på beslut till kommunfullmäktige')' (Hedemora); ' . . . via citizen proposal, you as a citizen in the municipality of Hudiksvall can have influence on that what happens in our municipality ('genom medborgarförslag kan du som medborgare i Hudiksvallkommun vara med och påverka det som händer i vår kommun')'. It can be added that the term for initiatives in Swedish is 'people'-based (folkinitiativ). In English versions of public content, both citizens' initiative and people's initiative are used. Even the Swedish term for the referendums is people-based (folkomröstning))

Citizenship can be seen as a socially constructed practice [70], and there is a growing interest in understanding citizenship's power as practice and status [137]. For example, Dominelli and Moosa-Mitha [70] examined how social workers in practice addressed issues of citizenship. In the field of participation in local politics, formal citizenship and national voting qualification are not relevant. Municipal officials acting as 'participatory engineers' [18] and 'democracy operators' [113] should therefore review their practice of citizenship as it can influence the integration of immigrants. Martiniello [21] identifies four dimensions of integrating immigrants in politics: acquiring rights; subjective identification with the host society; adopting democratic values; and, finally, political participation. Goodman and Wright [22] identified three stages of immigrant socio-economic and political integration. The first order achievement is language and/or knowledge acquisition. The second order achievement is functional navigation/meeting of immediate needs, for example navigating the health care system. The third order achievement is membership

in the polity, with civic navigation/identification with the polity. The participation of immigrants in the political process is a vital aspect of the integration of immigrants into their new society, enabling immigrants to become politically represented and to obtain political equality [138].

## 7. Conclusions

The empirical focus of this paper was whether political engagement at the local level is supported by using inclusive language and by clarifying online information about different means for participation.

The first research question was whether local governments provide clear information on means of participation between elections. Although the vast majority of their websites have a heading about politics, few of them provide comprehensive information about different means of participation, and the information about the instrument of 'direct democracy people's initiative' is exceptionally limited. Although SALAR [132] stresses that clear information about how to participate between elections is important for getting people involved in democracy at the local level, SALAR should also encourage municipalities to provide such information on their websites.

The second research question asked if SALAR is taking leadership for the adjustment of democracy to population changes in terms of who is and who is not a citizen. Although SALAR is generally a 'policy entrepreneur' in the field of local political participation [93], they still use the term *citizen*, both as an instrument for dialogue, and when targeting the public and the residents in the municipalities. In this way, SALAR undermines its own mission. Thus, instead of helping municipalities engage all of their residents, they contribute to marginalizing non-citizens who could otherwise actively contribute to local democracy.

The third and final research question concerned inclusiveness of terms used by local authorities on their websites. According to findings, local governments address the members of their municipalities in the context of political participation in different ways, for example as 'resident' or 'you'. Still, the term *citizen* is used by one third of municipalities. Furthermore, as an instrument for dialogue the term citizen dominates; there are examples of the use of 'resident dialogue', but they are few. In short, municipal leaders are also excluding residents who have the legal right to participate in municipal actions.

Due to migration, population structures are rapidly changing in many countries and a growing number of residents are non-citizens. As a consequence, the use of the term 'citizen' in the context of any public participation or civic engagement that does not require formal citizenship can be regarded as a social construction of exclusiveness. Local government authorities and agencies, as well other actors working with local political engagement, should thus be very careful about the term used both when explaining forms of participation and when addressing their public or individuals.

Based on a study of 700 citizen dialogue projects in Swedish municipalities, a summary publication [128] paints a nuanced picture of the pros and cons of resident involvement in local politics (outside the local government formal decision-making agenda). One general conclusion is that the dialogue is a baseline for residents to have a potential influence in specific matters of policy-making. This kind of dialogue arrangement represents a kind of 'mini-publics' [139]. In line with this conclusion, SALAR should review its own use of the term *citizen*, both when addressing Swedish residents—citizens or non-citizens in a formal sense—at the local level and in its work with participatory arrangements. This association of local and regional governments should also encourage its member municipalities to use inclusive terms on their local websites and portals, especially when informing residents about opportunities for local political engagement.

The study of civic-engagement-related information in this article is supply-oriented. Future research should rather focus on the demand side, i.e., the residents. Invited participation opportunities will not have any impact unless residents are aware of their existence. Future research should explore mechanisms of how residents acquire information about participatory means available in their municipalities. For example, future research should

investigate whether online municipal information is a way to increase residents' knowledge and engagement in local politics.

This article started with a citation that illustrates that non-citizen residents may feel excluded when citizen-based terms for participatory means are used. Future research should study if this feeling of exclusion is a common experience and explore whether non-exclusionary language makes any difference in terms of local people's interest in and influence on local politics. Sweden has 290 municipalities, and as demonstrated in the PLAN-study referred to above, there is great variation between dialogue projects, which issues are at stake, and how local decision-makers and invited citizens and other residents respond.

Participation could be individual or collective, legal or illegal, aiming at consensus or challenging law and order [140]. Thus, studying "mini-publics" in action must be embedded in a wider policy context and have a longer time horizon than one particular project [139] (p. 246). This complexity requires conceptually informed, in-depth studies of civic participation in specific projects and municipalities, including direct observation and interviews with 'democracy operators' [113]. Thereby, we can increase our understanding of how residents interact with municipalities such processes and give input to inspire future development of resident participation, dialogue, and influence in local politics.

**Funding:** This research received no external funding.

**Institutional Review Board Statement:** Not applicable.

**Informed Consent Statement:** Not applicable.

**Data Availability Statement:** Not applicable.

**Acknowledgments:** The author wish to thank the three anonymous reviewers for their valuable comments.

**Conflicts of Interest:** The author declares no conflict of interest.

## Appendix A

**Table A1.** Indicators for Transparency and Influence (Öppenhet och Påverkan) in the SALAR survey on municipal websites in Sweden, conducted yearly between 2009–2017.

| Issue | Accessibility "to" | Accessibility "How" |
|---|:---:|:---:|
| The complete budget | x | |
| A simplified version of budget adjusted for the citizens and target groups | x | |
| General information about how complaints and opinions are handled | x | |
| Handling of complaints and opinions | x | |
| Information on distribution of seats from the last election | x | |
| Information about coalition, alliance, and technical cooperation in elections | x | |
| Contact information for chairpersons of the municipal council, municipal executive board, and committees | x | |
| Information about the telephone number of all the politicians in the municipal council and on the committees | x | |
| Frequently asked questions (FAQs) are collected | x | |
| A search function and an A-Z index with municipalities' responsibility and contact information | x | |
| The complete annual report | x | |
| A simplified version of the annual report for the citizens of the municipality | x | |
| Possibility of subscribing to an electronic newsletter | x | |
| Information (or details of agenda, time, and place) about municipal council meetings | x | |
| Information (or details of agenda, time, and place) about municipal executive board meetings | x | |
| Information (or details of agenda, time, and place) about municipal committee meetings | x | |

**Table A1.** *Cont.*

| Issue | Accessibility "to" | Accessibility "How" |
|---|:---:|:---:|
| Documents for municipal council meetings before meetings have occurred | x | |
| Documents for municipal executive board meetings before meetings have occurred | x | |
| Documents for committee meetings before meetings have occurred | x | |
| Protocols of municipal council meetings | x | |
| Protocols of municipal executive board meetings | x | |
| Protocols of committee meetings | x | |
| Possibility for citizens to search in the municipality's records | x | |
| The website has been adapted so that it is easy to read | | x |
| The website has information in sign language | | x |
| Information about municipality activities can be found in languages other than Swedish (English) | | x |
| Municipal council meetings are distributed through Web TV | | x |
| Information about municipalities' insurance | x | |
| Use of social media on the Web (e.g., Facebook) | | x |

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
