# Peer review of "Only for Citizens? Local Political Engagement in Sweden and Inclusiveness of Terms"

_sustainability, doi:10.3390/su13147839_

Round 1

Reviewer 1 Report

Dear Author/s,

In my modest opinion, your article is very interesting as discussing the case of Sweden, however, it is unclear why the Sweden case is more unique than Finnish or Denmark. 

From another perspective, you are trying to connect your article with the content of the journal by mentioning lines 31 and 32 a few times the word "sustainability".  And that's all that we know about sustainability and citizen's participation in Sweden.  In my modest opinion, you must resubmit your article to other journals of the MDPI like Administrative Sciences or Social Sciences or Societies as your article and contribution very promising in the case of these journals. 

As of now, there is no direct link between sustainability and citizens participation. 

Thank you for understanding & have a good day!

ps. I also recommend adding the sentence on the methodology in the abstract

Author Response

Dear Revewer 1, I am grateful to your insightful comments on my paper.

Comment: As of now, there is no direct link between sustainability and citizens participation. 
Response: Thank you for pointing this out. Per your suggestion, the second paragraph in the Introduction now explains that civic engagement and participation is commonly considered as crucial for achieving sustainability and for successfully monitoring the Sustainable Development Goals.

Comment: ps. I also recommend adding the sentence on the methodology in the abstract
Response: I agree. In the Abstract there is now some brief information about how 290 websites have been studied.  

Reviewer 2 Report

The paper is an interesting contribution in the field of political engagement, deepening into a particular case in Sweden. In terms of quality, the manuscript is well structured, well written, and well systematized. There are only two places which require revision. 1) In the introduction, the authors list the three research questions. While before this list, they review the literature and somehow make the case to justify their questions, it would be better to frame every question in the scientific literature and showing how each come to cover a gap. 2) In the conclusions, the authors should include limitations of the study as well as point out future lines of research. Finally, the quality of figure 1 is not very good. It would be better to replace it for one of more quality.

Author Response

Dear Reviewer 2, I am grateful to your insightful comments on my paper.

Comment: In the introduction, the authors list the three research questions. While before this list, they review the literature and somehow make the case to justify their questions, it would be better to frame every question in the scientific literature and showing how each come to cover a gap.
Response: I agree. The research questions are now presented together after problematization with the relevant literature.

Comment: In the conclusions, the authors should include limitations of the study as well as point out future lines of research.
Response: Thank you for pointing this out.  Limitations of the study and lines for future research are discussed in three paragraphs.

Comment: Finally, the quality of figure 1 is not very good. It would be better to replace it with one of more quality. 
Response: I agree. The quality of figure 1 has been improved.

Reviewer 3 Report

Thanks. I found the study interesting, relevant, up-to-date, well-written and focused. 

Author Response

Dear reviewer 3, I was happy to get your response that article was interesting, relevant and focused.

Reviewer 4 Report

This is well written and interesting paper showing participation in a very interesting (migration) context. The extend of the text proves that the author(s) pay(s) attention in both theoretical and methodological part of the research. I have read the text with huge interest and recommend it anyone who is interested in this topis. However, I would like to suggest one amendment to insert into the first part of the paper. It would be very useful to show the concept of the research on a chart/ graph. I may help to grasp the whole idea of the research more easily. 

Moreover, given the fact that conducted research has been based on websites and day-to-day sources I would suggest to provide some recommendations for local government associations or local governments. 

All comments mantioned above does not change the fact that the paper is very interesting and should be published anyway.  

Author Response

Dear Reviewer 4, I am grateful to your insightful comments on my paper.

Comment: I would like to suggest one amendment to insert into the first part of the paper. It would be very useful to show the concept of the research on a chart/ graph. It may help to grasp the whole idea of the research more easily. 
Response: Thank you for this suggestion. I hope that the introduction section, methodology description and the clear structure of the article contribute to a good understanding of the paper.

Comment: Moreover, given the fact that conducted research has been based on websites and day-to-day sources I would suggest to provide some recommendations for local government associations or local governments.
Response: Some policy-related recommendations were already in place. These have been made more clear and a paragraph has been added.

Round 2

Reviewer 1 Report

As I have mentioned in my review the paper is good and it must to be consider for the publication in any other MDPI journals but it is still has not a lot of connection with Sustainability journal.